# Synthesis of a Small Library of Nature-Inspired Xanthones and Study of Their Antimicrobial Activity

**DOI:** 10.3390/molecules25102405

**Published:** 2020-05-21

**Authors:** Diana I. S. P. Resende, Patrícia Pereira-Terra, Joana Moreira, Joana Freitas-Silva, Agostinho Lemos, Luís Gales, Eugénia Pinto, Maria Emília de Sousa, Paulo Martins da Costa, Madalena M. M. Pinto

**Affiliations:** 1CIIMAR - Centro Interdisciplinar de Investigação Marinha e Ambiental, Terminal de Cruzeiros do Porto de Leixões, 4450-208 Matosinhos, Portugal; dresende@ff.up.pt (D.I.S.P.R.); tichaterra@gmail.com (P.P.-T.); joana.m26@hotmail.com (J.M.); joanafreitasdasilva@gmail.com (J.F.-S.); pmcosta@icbas.up.pt (P.M.d.C.); madalena@ff.up.pt (M.M.M.P.); 2Laboratório de Química Orgânica e Farmacêutica, Faculdade de Farmácia, Universidade do Porto, Rua de Jorge Viterbo Ferreira 228, 4050-313 Porto, Portugal; up201002662@ff.up.pt; 3ICBAS – Instituto de Ciências Biomédicas Abel Salazar, Universidade do Porto, Rua de Jorge Viterbo Ferreira 228, 4050-313 Porto, Portugal; lmgales@gmail.com; 4i3S – Instituto de Investigação e Inovação em Saúde, Rua Alfredo Allen 208, 4200-135 Porto, Portugal; 5IBMC – Instituto de Biologia Molecular e Celular Universidade do Porto, Rua Alfredo Allen 208, 4200-135 Porto, Portugal; 6Laboratório de Microbiologia, Departamento de Ciências Biológicas, Faculdade de Farmácia, Universidade do Porto, Rua de Jorge Viterbo Ferreira 228, 4050-313 Porto, Portugal

**Keywords:** xanthones, diversity-oriented synthesis, antifungal activity, antibacterial activity

## Abstract

A series of thirteen xanthones **3**–**15** was prepared based on substitutional (appendage) diversity reactions. The series was structurally characterized based on their spectral data and HRMS, and the structures of xanthone derivatives **1**, **7**, and **8** were determined by single-crystal X-ray diffraction. This series, along with an in-house series of aminated xanthones **16**–**33,** was tested for in-vitro antimicrobial activity against seven bacterial (including two multidrug-resistant) strains and five fungal strains. 1-(Dibromomethyl)-3,4-dimethoxy-9*H*-xanthen-9-one (**7**) and 1-(dibromomethyl)-3,4,6-trimethoxy-9*H*-xanthen-9-one (**8**) exhibited antibacterial activity against all tested strains. In addition, 3,4-dihydroxy-1-methyl-9*H*-xanthen-9-one (**3**) revealed a potent inhibitory effect on the growth of dermatophyte clinical strains (*T. rubrum* FF5, *M. canis* FF1 and *E. floccosum* FF9), with a MIC of 16 µg/mL for all the tested strains. Compounds **3** and **26** showed a potent inhibitory effect on two *C. albicans* virulence factors: germ tube and biofilm formation.

## 1. Introduction

Multi-drug resistance is one of the major causes of the alarming level of infectious disease worldwide, with treatment failure being an increasing concern. The discovery of new antimicrobial drugs, which can overcome problems of resistance to current anti-infective drug therapies, is urgent, and requires efforts in industry and scientific research communities. Natural products have been the most successful source of potential drug leads for millennia, and the influence of natural product structures is nowadays quite marked in the anti-infective area, most related to their role in defense mechanisms of secondary metabolites. Xanthones are ubiquitous polyphenolic secondary metabolites, and, particularly, several naturally occurring xanthones have revealed potent antimicrobial activities (Figure 1) [1]. For example, norlichexanthone [2,3,4,5], 1,4,5-trihydroxy-2-methylxanthone [6], fischexanthone [7], and dimethyl 8-methoxy-9-oxo-9*H*-xanthene-1,6-dicarboxylate [8,9] (Figure 1) have revealed potent antimicrobial activities against several bacterial and fungal strains. Interestingly, some xanthone precursors, like 3,5-dibromo-2-(2,4-dibromophenoxy) phenol and 3,4,5-tribromo-2-(2,4-dibromophenoxy)phenol (Figure 1), also revealed potent antibacterial activity against Gram-positive and Gram-negative bacteria [10]. The dibenzo-gamma-pyrone scaffold is considered a privileged structure due to the ability of different derivatives to display different biological activities. Although a myriad of substitution patterns can be found in natural xanthones, the presence of certain groups in specific positions imposed by their biosynthetic pathway is a known limitation that can be surpassed by the use of organic synthesis [11]. Therefore, this scaffold has been an interesting starting point for the discovery of new potential drug candidates due to its ability to display binding functionalities from a rigid dibenzo-gamma-pyrone core [12,13,14,15,16,17,18]. In a recent work, our group described the synthesis of a series of novel nature-inspired chlorinated xanthones and their antimicrobial activity [19]. The promising results prompted us to explore the versatility of this scaffold with simple chemical transformations, starting from xanthones with a 3,4-dioxygenated pattern of substitution based on previous structure–activity relationship studies (SAR) and employing different reagents in order to achieve a library diversity in terms of molecular function. Representative substituents inspired by nature (Figure 1: carboxylic acid, ester, methyl, methoxyl, phenol, bromo substituents) were selected to obtain a library of antimicrobial xanthones.

## 2. Results and Discussion

### 2.1. Chemistry

In order to synthetize a collection of structurally diverse compounds, several straightforward transformations were performed for two simple oxygenated xanthones: 3,4-dimethoxy-1-methyl-9*H*-xanthen-9-one (**1**) and 3,4,6-trimethoxy-1-methyl-9*H*-xanthen-9-one (**2**) (Scheme 1), obtained as previously described [19,20]. The corresponding phenols, **3** and **4,** were prepared using AlCl_3_ as the *O*-demethylating agent. Selective C-2 bromination of **1** and **2** with a Bu_4_NBr/PhI(OAc)_2_ system under mild conditions produced 2-bromo-3,4-dimethoxy-1-methyl-9*H*-xanthen-9-one (**5**) and 2-bromo-3,4,6-trimethoxy-1-methyl-9*H*-xanthen-9-one (**6**) [21]. The low yields obtained in this reaction are due to failure of the reaction to go to completion and other complications in the purification step. On the other hand, Wohl–Ziegler bromination of **1** and **2** using *N*-bromosuccinimide (NBS) brominating agent and benzoyl peroxide (BPO) as the radical initiator afforded 1-(dibromomethyl)-3,4-dimethoxy-9*H*-xanthen-9-one (**7**) [22] and 1-(dibromomethyl)-3,4,6-trimethoxy-9*H*-xanthen-9-one (**8**), respectively. Subsequent solvolytic displacement of bromine atoms of the *gem*-dibromomethylated derivatives **7** and **8** was successfully accomplished using 1-butyl-3-methylimidazolium tetrafluoroborate ((bmIm)BF_4_) and water, under conventional heating, and furnished carbaldehydic xanthones **9** [22] and **10** in good yields.

To further extend the diversity of the compound library, and due to the high versatility of the formyl group in organic chemistry, additional transformations were performed in carbaldehydic xanthones **9** and **10** (Scheme 2). Oxidation employing Oxone^®^ as the sole oxidant [23] to the corresponding carboxylic acids **11** and **12** and to ester products **13** and **14** was effectively accomplished in good yields either for the acids (71% and 68%, respectively) or the corresponding esters (53% and 77%, respectively). Furthermore, a condensation of xanthone **9** with hydroxylamine made it possible to obtain aldoxime **15** with a moderate yield (21%), justified by laborious purification and consequent product losses.

### 2.2. Structure Elucidation

The structure of the new xanthone derivatives **3**–**6**, **8**, **10** and **12**–**15** was established by ^1^H- (Table 1) and ^13^C- (Table 2) nuclear magnetic resonance (NMR), and high-resolution mass spectrometry (HRMS) techniques (Appendix A). The ^13^C-NMR assignments were determined by bidimensional heteronuclear single quantum correlation (HSQC) and heteronuclear multiple bond correlation (HMBC) experiments. The assignments of carbon atoms directly bonded to proton atoms were achieved from HSQC experiments, and the chemical shifts of carbon atoms not directly bonded to proton atoms were deduced from HMBC correlations.

The structural elucidation of compounds **1**, **2**, **7**, **9** and **11** was established by comparing their ^1^H and ^13^C-NMR data with those reported in the literature [19,20]. In general, ^1^H-NMR spectra of the synthesized xanthones **3**–**6**, **8**, **10**, **12**–**15** (Table 1) show the presence of five (or four if 6-substituted) signals corresponding to the aromatic protons, namely H-2, H-5, (H-6), H-7, and H-8 (δ_H_ 6.84–8.43 ppm). The aromatic protons most deshielded are the protons H-6 and H-8. The electron-withdrawing effect of carbonyl group (C-9) for resonance contributes to an electronic deprotection of ortho (H-8) and para (H-6) positions of the aromatic ring. The magnetic anisotropy of carbonyl group (C-9) preferentially deshields the proton H-8, which explains the higher chemical shift of proton H-8 when compared to H-6. Additionally, the protons corresponding to the methoxyl groups [H-3, H-4 (and H-6)] are equivalent for each methoxyl group. In spite of their similarity, the H-4 protons present a slightly higher chemical shift than the H-3 protons. This evidence was confirmed by heteronuclear single quantum correlation (HSQC) and heteronuclear multiple bond correlation (HMBC) techniques. The ^13^C-NMR spectra of the synthesized xanthones **3**–**6**, **8**, **10**, **12**–**15** (Table 2) revealed the presence of a highly deshielded signal corresponding to the resonance of the carbonyl carbon (δ_C-9_ 176.0–178.0 ppm). It was also possible to visualize signals corresponding to the remaining twelve carbons of the xanthone scaffold (δ_C_ 99.7–165.5 ppm). The two (or three) signals corresponding to the resonance of the protons of the methoxyl groups [3-OCH_3_, 4-OCH_3_ (and 6-OCH_3_)] and one regarding the resonance of the protons of the methyl group (C-1′) presented the lowest chemical shifts. ^13^C-NMR assignments of directly bound nuclei were determined by HSQC and, on the other hand, nuclei separated from each other with two or more chemical bonds were deduced by HMBC experiments.

As an example, the main correlations between protons and carbons for 1-(dibromomethyl)-3,4,6-trimethoxy-9*H*-xanthen-9-one **8** are represented in Figure 2. The highly deshielded H-1′ (δ_H-1′_ 8.98 ppm) can be identified through the correlation with C-2 and C-9a. Between the aromatic protons, H-8 (δ_H-8_ 8.20 ppm) appears in the form of a doublet and presents a correlation with C-6, C-9, and C-10a. On the other hand, the signals corresponding to the resonance of protons H-2 (δ_H-2_ 7.74 ppm) and H-5 (δ_H-5_ 6.97–6.92 ppm) are easily distinguished through the correlations presented: while H-2 correlates with C-1′, C-3, C-4 and C-9a, H-5 presents correlations with C-6, C-7, C-8a, and C-10a.

The structures of three xanthone derivatives, **1**, **7**, and **8,** were determined by single-crystal X-ray diffraction and are shown in Figure 3. The xanthone skeleton adopts a flattened boat conformation in the three structures, as usual [24]. The 3- and 6- methoxyl substituents adopt a coplanar conformation relative to the three-ring system, allowing maximum overlap of the unshared oxygen electrons with the aromatic π electron cloud. This conformation creates a close approach between the methoxyl carbon and the adjacent aromatic carbon. On the other hand, the 4-methoxy substituent is hindered by the adjacent substituent, and the carbon atom of OCH_3_ is well out of the aromatic plane due to a rotation along the C-OCH_3_ bond.

### 2.3. Microbiology

All of the 11 synthesized xanthones **3–15**, along with a series of 18 in-house 3,4-dioxygenated xanthones **16–33** (Table 3) previously synthetized were tested for their in-vitro antimicrobial activity against five fungal (Table 4) and seven bacterial (including two multidrug-resistant) strains (Table 5).

The results for the antifungal activity of the tested compounds against yeast and filamentous fungi that exhibited activity are presented in Table 4. None of the compounds tested showed activity against *C. albicans* nor *A. fumigatus* strains except **24,** with a high MIC of 128 µg/mL for *C. albicans*.

Nevertheless, compounds **3**, **4**, **13**, **23**, **26**, **27** and **31** revealed a variable inhibitory effect on dermatophytes, with a MIC and MFC ranging from 16 to 128 µg/mL, depending on the compounds and the species tested. Compounds **3** and **26** revealed a potent inhibitory effect on the growth of dermatophyte clinical strains (*T. rubrum* FF5, *M. canis* FF1 and *E. floccosum* FF9), with a MIC of 16 µg/mL for **3** and 32 µg/mL for **26** for all the tested strains. The fungicidal activity at the same concentration at that MIC was observed only for compound **3** with *M. canis* FF1 and compound **27** with *T. rubrum* FF5.

Compounds **3**, **4**, **13**, **23**, **26**, **27** and **31** were also evaluated for synergistic effects for *T. rubrum*. No synergy was observed with fluconazole (data not shown).

For compounds with some antifungal activity (Table 4), their effect on processes associated with *C. albicans* virulence was assessed, namely on germ tube and biofilm formation. Only **3** and **26** had an inhibitory effect on germ tube formation of *C. albicans* ATCC 10,231 (Figure 4), with no germination at 128 μg/mL and 64 μg/mL and a significant inhibition at 32 μg/mL, even though these compounds had no effect on overall growth at these concentrations.

For **3** and **26**, antibiofilm activity was also evaluated; as germ tube formation plays a key role in biofilm formation, it is one of the major virulence factors contributing to the pathogenesis of candidiasis [26]. In comparison to an untreated control, **26** significantly impaired biofilm formation of *C. albicans* ATCC 10,231 at all concentrations tested (128–16 μg/mL), while **3** had a significant impact at concentrations ranging between 128 and 32 μg/mL (Figure 5). Minimum biofilm inhibitory concentration (MBIC), which is defined as the minimum compound concentration that leads to an 80% reduction of biofilm formation compared to an untreated control [27], was 32 μg/mL for **26** and > 128 μg/mL for **3**. Nonetheless, these properties of **3** and **26** should be studied further, including their potential to be associated with existing antifungals.

In order to evaluate the antimicrobial activity of compounds **3–5**, **7**, **8** and **10–33** against Gram-positive and Gram-negative bacteria, an initial activity screening was performed by the disk diffusion method for several reference strains and environmental multidrug-resistant isolates. The results of the active compounds are presented in Table 5. Compounds **7**, **8**, **20** and **26** revealed antibacterial activity against Gram-negative bacteria, producing a halo of inhibition of 8, 8, 8 and 10 mm for *E. coli* ATCC 25922, respectively. Regarding *P. aeruginosa* ATCC 27853, none of the compounds were able to generate a visible zone of inhibition, with the exception of compound **8,** which displayed an inhibitory halo with 8 mm in diameter. Moreover, those compounds were also capable of inhibiting the growth of an ESBL *E. coli* strain (SA/2), ensuing a similar inhibition zone to that of the reference strain (9, 8, 9, 8 and 9.5 mm respectively). Regarding Gram-positive bacteria, compounds **7**, **8**, **20** and **26** displayed an inhibitory effect against *E. faecalis* ATCC 29212, with inhibition halos of 9, 10, 9, 9 and 8.5 mm, whereas compounds **7**, **8**, **26** and **27** were active against *S. aureus* ATCC 29,213 with inhibition halos of 9, 11, 9 and 9.5 mm, respectively. Similarly, **7**, **8**, and **26** inhibited the growth of either methicillin-resistant *S. aureus* (MRSA) or vancomycin-resistant Enterococci (VRE), resulting in an inhibition zone of 10, 11 and 9 mm for MRSA and 9, 8 and 8.5 mm for VRE.

Additionally, **12** inhibited MRSA growth, presenting an inhibition halo of 8 mm, and compound **20** inhibited VRE, with an inhibition halo of 9 mm. Despite these encouraging results, it was not possible to determine an MIC for any compound in any of the strains in the range of concentrations tested. This might be related to the fact that some compounds are poorly soluble in the culture media used for the determination of the MIC, and the amount of available compound in the solution was probably lower than intended. Regarding the screening for potential synergies with multidrug-resistant bacterial strains and the tested compounds in combination with clinically relevant antibiotics, none of the compounds revealed a synergistic association with antibiotics (data not shown).

Concerning SAR analysis, the obtained results were consistent with data previously reported for some natural [1,28,29] and synthetic xanthones [19], the halogenated derivatives being in general the most promising antimicrobial agents (Figure 6). For antifungal activity, the presence of only two hydroxyl groups (**3**) at C-3 and C-4 seems to be important, since compound **4**, with an additional hydroxyl group at C-6, was not as potent as **3**; on the other hand, the presence of activating groups (CH_3_) at C-1 (**3**) is more favorable than the presence of deactivating groups (COOCH_3_) at the same position (**13**). Moreover, we observed some tolerance to variations in the substituent groups at C-1 position, namely amine moieties (**23**, **24**, **26**, **27**, **31**), with aminated xanthones with halogenated aromatic rings being the most promising concerning antifungal activity (**26, 27**) against all the tested dermatophyte clinical strains (*T. rubrum* FF5, *M. canis* FF1 and *E. floccosum* FF9). Regarding antibacterial activity, SAR suggests that the presence of two bromine atoms (**7** and **8**) plays an important role towards this activity. Interestingly, compounds with a halogen atom at the amine moiety (**20**, **26, 27**) exhibited potent antibacterial activity, suggesting once more that the halogen atoms are important for activity.

## 3. Materials and Methods

### 3.1. Chemistry

#### 3.1.1. Materials and General Methods

All reagents and solvents were purchased from TCI (Tokyo Chemical Industry Co. Ltd., Chuo-ku, Tokyo, Japan), Acros (Geel, Belgium), Sigma Aldrich (Sigma-Aldrich Co. Ltd., Gillingham, UK), or Alfa Aesar (Thermo Fisher GmbH, Kandel, Germany) and had no further purification process. Solvents were evaporated using a rotary evaporator (BÜCHI Labortechnik AG, Flawil, Switzerland) under reduced pressure, Buchi Waterchath B-480. Microwave (MW) reactions were performed using an Ethos MicroSYNTH 1600 Microwave Labstation from Milestone (Thermo Unicam, Waltham, MA, USA). The internal reaction temperature was controlled by a fiber-optic probe sensor. All reactions were monitored by TLC carried out on precoated plates with 0.2-mm thickness using Merck silica gel 60 (GF254) with appropriate mobile phases and detection at 254 and/or 365 nm. Purification of the synthesized compounds was performed by chromatography flash column using Merck silica gel 60 (0.040–0.063 mm). Melting points (m.p.) were measured in a Köfler microscope (Wagner and Munz, Munich, Germany) and are uncorrected. ^1^H- and ^13^C-NMR spectra were taken in CDCl_3_ at room temperature on a Bruker Avance 300 instrument (Bruker Biosciences Corporation, Billerica, MA, USA) (300.13 or 500.13 MHz for ^1^H- and 75.47 or 125.77 MHz for ^13^C-). Chemical shifts are expressed in δ (ppm) values relative to tetramethylsilane (TMS) as an internal reference. Coupling constants are reported in hertz (Hz). ^13^C-NMR assignments were made by 2D HSQC and HMBC experiments (long-range C, H coupling constants were optimized to 7 and 1 Hz). HRMS mass spectra were measured on a Bruker Daltonics micrOTOF Mass Spectrometer (Bruker Corporation, Billerica, MA, USA), recording in ESI (electrospray) mode in Centro de Apoio Científico e Tecnolóxico á Investigation (C.A.C.T.I.), University of Vigo, Galicia, Spain.

#### 3.1.2. General Procedure for the Synthesis of 3,4-Dihydroxy-1-methyl-9*H*-xanthen-9-one (**3**) and 3,4,6-Trihydroxy-1-methyl-9*H*-xanthen-9-one (**4**)

The appropriate methoxy-1-methyl-9*H*-xanthen-9-one (**1** or **2**, 0.666 mmol) was dissolved in dry toluene (15 mL), and aluminum chloride (6.660 mmol) was carefully added. The reaction mixture was refluxed, with magnetic stirring under a nitrogen atmosphere, for 1.5 h. After cooling the reaction mixture to room temperature, excess conc. HCl (5N, 10 mL) was added and the mixture was extracted with ethyl acetate (4 × 50 mL) and the organic layers were evaporated and dried with Na_2_SO_4_ and the solvent was removed under reduced pressure to obtain the crude product. Purification by preparative TLC (SiO_2_, chloroform/methanol 9:1) gave the pure products 3,4-dihydroxy-1-methyl-9*H*-xanthen-9-one (**3**, 51.4 mg, 30% yield) or 3,4,6-trihydroxy-1-methyl-9*H*-xanthen-9-one (**4**, 63.7 mg, 37% yield) as brown solids.

*3,4-Dihydroxy-1-methyl-9H-xanthen-9-one* (**3**); Brown solid (51.4 mg, 30% yield); m.p. = 265–267 °C. ^1^H-NMR (DMSO-*d*_6_, 300.13 MHz): δ = 10.34 (s, 1H, 3-OH), 9.18 (s, 1H, 4-OH), 8.11 (dd, ^3^*J*_8,7_ = 8.0 Hz, ^4^*J*_8,6_ = 1.8 Hz, 1H, H-8), 7.78 (ddd, ^3^*J*_6,5_ = 8.6 Hz, ^3^*J*_6,7_ = 7.1 Hz, ^4^*J*_6,8_ = 1.8 Hz, 1H, H-6), 7.57 (dd, ^3^*J*_5,6_ = 8.6 Hz, ^4^*J*_5,7_ = 1.1 Hz, 1H, H-5), 7.40 (ddd, ^3^*J*_7,8_ = 8.0 Hz, ^3^*J*_7,6_ = 7.1 Hz, ^4^*J*_7,5_ =1.1 Hz, 1H, H-7), 6.68 (s, 1H, H-2), 2.68 (s, 3H, 1-CH_3_) ppm. ^13^C-NMR (DMSO-d_6_, 75.47 MHz): δ = 176.9 (C-9), 154.7 (C-10a), 150.4 (C-3), 147.5 (C-4a), 134.4 (C-6), 131.2 (C-1), 130.7 (C-4), 126.0 (C-8), 123.7 (C-7), 121.8 (C-8a), 117.5 (C-5), 115.2 (C-2), 112.8 (C-9a), 22.4 (1-CH_3_) ppm. HRMS (ESI^+^): *m*/*z* [C_14_H_10_O_4_ + H]^+^ calcd. for [C_14_H_11_O_4_]: 243.06519; found 243.06505.

*3,4,6-Trihydroxy-1-methyl-9H-xanthen-9-one* (**4**); Brown solid (63.7 mg, 37% yield); m.p. >300 °C. ^1^H-NMR (DMSO-d_6_, 300.13 MHz): δ = 7.93 (d, ^3^*J*_8,7_ = 9.1 Hz, 1H, H-8), 6.84–6.77 (m, 2H, H-5, H-7), 6.62 (s, 1H, H-2), 2.65 (3H, s, 1-CH_3_) ppm. ^13^C-NMR (DMSO-d_6_, 75.47 MHz): δ = 176.2 (C-9), 162.9 (C-6), 156.5 (C-10a), 149.8 (C-3), 147.4 (C-4a), 130.7 (C-1, C-4), 127.9 (C-8), 114.9 (C-2, C-8a), 113.3 (C-7), 112.6 (C-9a), 101.6 (C-5), 22.4 (1-CH_3_) ppm. HRMS (ESI^+^): *m*/*z* [C_14_H_10_O_5_ + H]^+^ calcd. for [C_14_H_11_O_5_]: 259.06010; found 259.05987.

#### 3.1.3. General Procedure for the Synthesis of 2-Bromo-3,4-dimethoxy-1-methyl-9*H*-xanthen-9-one (**5**) and 2-Bromo-3,4,6-trimethoxy-1-methyl-9*H*-xanthen-9-one (**6**)

PhI(OAc)_2_ (1.110 mmol) was suspended in anhydrous CH_2_Cl_2_ (2 mL) under a nitrogen atmosphere at room temperature. Bu_4_NBr (1.110 mmol) was added and the mixture was stirred at room temperature for 30 min. The appropriate methoxy-1-methyl-9*H*-xanthen-9-one (0.370 mmol) in anhydrous in CH_2_Cl_2_ (2 mL) was added and the mixture was stirred at 40 °C for 7 days. The reaction mixture was quenched with saturated aqueous ammonium chloride, and the aqueous portion was separated and extracted with CH_2_Cl_2_. The organic layer was dried over MgSO_4_ and evaporated in vacuo. The crude product was purified by preparative TLC (SiO_2_, EtOAc/hexane 2:8) to give the pure products 2-bromo-3,4-dimethoxy-1-methyl-9*H*-xanthen-9-one (**5**, 49.6 mg, 38% yield) or 2-bromo-3,4,6-trimethoxy-1-methyl-9*H*-xanthen-9-one (**6**, 9.0 mg, 7% yield) as white solid.

*2-Bromo-3,4-dimethoxy-1-methyl-9H-xanthen-9-one* (**5**); White solid (49.6 mg, 38% yield); m.p. 171–173 °C. ^1^H-NMR (CDCl_3_, 300.13 MHz): δ = 8.27 (ddd, ^3^*J*_8,7_ = 8.1 Hz, ^4^*J*_8,6_ = 1.7 Hz, ^5^*J*_8,5_ = 0.5 Hz, 1H, H-8), 7.70 (ddd, ^3^*J*_6,5_ = 8.6 Hz, ^3^*J*_6,7_ = 7.1 Hz, ^4^*J*_6,8_ = 1.7 Hz, 1H, H-6), 7.52 (dd, ^3^*J*_5,6_ = 8.6 Hz, ^4^*J*_5,7_ = 1.1 Hz, ^5^*J*_5,8_ = 0.5 Hz, 1H, H-5), 7.37 (ddd, ^3^*J*_7,8_ = 8.1 Hz, ^3^*J*_7,6_ = 7.1 Hz, ^4^*J*_7,5_ = 1.1 Hz, 1H, H-7), 4.07 (3H, s, 4-OCH_3_), 4.04 (3H, s, 3-OCH_3_), 3.05 (s, 3H, 1-CH_3_) ppm.^13^C-NMR (CDCl_3_, 75.47 MHz): δ = 177.8 (C-9), 154.9 (C-10a), 154.3 (C-3), 151.6 (C-4a), 139.8 (C-4), 137.0 (C-1), 134.6 (C-6), 127.1 (C-8), 124.4 (C-7), 122.7 (C-8a), 118.0, 117.8 (C-2, C-9a), 117.6 (C-5), 62.0 (3-OCH_3_), 61.4 (4-OCH_3_), 21.7 (1-CH_3_) ppm. HRMS (ESI^+^): *m*/*z* [C_16_H_13_BrO_4_ + H]^+^ calcd. for [C_16_H_14_BrO_4_]: 349.00700; found 349.00692.

*2-Bromo-3,4,6-trimethoxy-1-methyl-9H-xanthen-9-one* (**6**); White solid (9.0 mg, 7% yield); m.p. 171–173 °C. ^1^H-NMR (CDCl_3_, 300.13 MHz): δ = 8.18 (d, ^3^*J*_8,7_ = 8.8 Hz, 1H, H-8), 6.94 (dd, ^3^*J*_7,8_ = 8.8 Hz, ^4^*J*_7,5_ = 2.4 Hz, 1H, H-7), 7.52 (d, ^4^*J*_5,7_ = 2.4 Hz, 1H, H-5), 4.06 (s, 3H, 3-OCH_3_), 4.03 (3H, s, 4-OCH_3_), 3.94 (3H, s, 6-OCH_3_), 3.06 (s, 3H, 1-CH_3_) ppm. ^13^C-NMR (CDCl_3_, 75.47 MHz): δ = 176.9 (C-9), 164.8 (C-6), 156.6 (C-10a), 153.8 (C-3), 151.4 (C-4a), 139.7 (C-4), 136.8 (C-1), 134.6 (C-6), 128.5 (C-8), 118.0 (C-9a), 117.7 (8a), 116.5 (C-2), 113.6 (C-7), 99.7 (C-5), 61.9 (3-OCH_3_), 61.2 (4-OCH_3_), 55.9 (6-OCH_3_), 21.5 (1-CH_3_) ppm. HRMS (ESI^+^): *m*/*z* [C_17_H_15_BrO_5_ + H]^+^ calcd. for [C_17_H_16_BrO_5_]: 379.0176; found 379.0161.

#### 3.1.4. Synthesis of 1-(Dibromomethyl)-3,4-dimethoxy-9*H*-xanthen-9-one (**7**) and 1-(Dibromomethyl)-3,4,6-trimethoxy-9*H*-xanthen-9-one (**8**)

*1-(Dibromomethyl)-3,4-dimethoxy-9H-xanthen-9-one* (**7**) (2.46 g, 78%) was synthesized from 3,4-dimethoxy-1-methyl-9H-xanthen-9-one (**1**) and characterized according to the previously described procedures [22].

*1-(Dibromomethyl)-3,4,6-trimethoxy-9H-xanthen-9-one* (**8**). A mixture of 3,4,6-trimethoxy-1-methyl-9*H*-xanthen-9-one (0.629 g, 2.095 mmol), *N*-bromosuccinimide (0.746 g, 4.190 mmol) and dibenzoylperoxide (0.152 g, 0.628 mmol) in carbon tetrachloride (12 mL) was refluxed for 2 h under light (300 W). After cooling at 0 °C and stirring for 2 h, the precipitate was filtered and washed with cold carbon tetrachloride. The mother liquor was evaporated under reduced pressure and purified by flash chromatography (silica gel, petroleum ether/ethyl acetate 9:1) to obtain the pure product 1-(dibromomethyl)-3,4,6-trimethoxy-9*H*-xanthen-9-one (**8**, 0.320 g, 72% yield) as white needles (0.71 g, 74%); m.p. 159–161 °C. ^1^H-NMR (CDCl_3_, 300.13 MHz): δ = 8.98 (s, 1H, H-1′), 8.20 (d, ^3^*J*_8,7_ = 7.9 Hz, ^4^*J*_8,5_ = 1.3 Hz, 1H, H-8), 7.74 (1H, s, H-2), 6.97–6.92 (m, 2H, H-5, H-7), 4.10 (3H, s, 3-OCH_3_), 4.04 (3H, s, 4-OCH_3_), 3.95 (3H, s, 6-OCH_3_) ppm. ^13^C-NMR (CDCl_3_, 75.47 MHz): δ = 177.1 (C-9), 165.4 (C-6), 157.0 (C-10a), 156.3 (C-3), 150.5 (C-4a), 139.6 (C-1), 137.6 (C-4), 128.6 (C-8), 116.1 (C-8a), 114.0 (C-7), 111.9 (C-2), 111.0 (C-9a), 99.8 (C-5), 61.8 (4-OCH_3_), 56.7 (3-OCH_3_), 56.1 (6-OCH_3_), 39.5 (C-1′) ppm. HRMS (ESI^+^): *m*/*z* [C_17_H_14_Br_2_O_5_ + H]^+^ calcd. for [C_17_H_15_Br_2_O_5_]: 456.9281; found 456.9275.

#### 3.1.5. Synthesis of 3,4-Dimethoxy-9-oxo-9*H*-xanthene-1-carbaldehyde (**9**) and 3,4,6-trimethoxy-9-oxo-9*H*-xanthene-1-carbaldehyde (**10**)

*3,4-Dimethoxy-9-oxo-9H-xanthene-1-carbaldehyde* (**9**) (1.33 g, 64% yield) was synthesized from 1-(dibromomethyl)-3,4-dimethoxy-9*H*-xanthen-9-one (**7**) and characterized according to the previously described procedure [22].

*3,4,6-Trimethoxy-9-oxo-9H-xanthene-1-carbaldehyde* (**10**). A mixture of 1-(dibromomethyl)-3,4,6-trimethoxy-9*H*-xanthen-9-one (0.645 g, 1.408 mmol) and ionic liquid, 1-butyl-3-methylimidazolium tetrafluoroborate, [(bmim)BF_4_] (1.58 mL, 5:1) mixed with H_2_O, was heated at 100 °C with stirring for 3 h. The reaction mixture was allowed to cool and extracted with ethyl acetate (3 × 5 mL). The organic layer was dried with anhydrous sodium sulfate, concentrated under reduced pressure, and the crude material was purified by flash column chromatography (silica gel, petroleum ether/ethyl acetate 9:1) to give the pure product 3,4,6-trimethoxy-9-oxo-9*H*-xanthene-1-carbaldehyde (**10**, 0.320 g, 72%) as a fluffy white solid; m.p. 221–223 °C. ^1^H-NMR (CDCl_3_, 300.13 MHz): δ = 11.23 (s, 1H, CHO), 8.20 (d, ^3^*J*_8,7_ = 9.4 Hz, 1H, H-8), 7.52 (s, 1H, H-2), 6.99–6.94 (m, 2H, H-5, H-7), 4.09 (3H, s, 4-OCH_3_), 4.05 (3H, s, 3-OCH_3_), 3.95 (s, 3H, 6-OCH_3_) ppm. ^13^C-NMR (CDCl_3_, 75.47 MHz): δ = 193.1 (CHO), 177.1 (C-9), 165.5 (C-6), 157.5 (C-10a), 156.0 (C-3), 150.9 (C-4a), 140.7 (C-4), 133.6 (C-1), 128.3 (C-8), 116.4 (C-9a), 115.9 (C-8a), 114.2 (C-7), 108.3 (C-2), 100.1 (C-5), 61.9 (4-OCH_3_), 56.7 (3-OCH_3_), 56.1 (6-OCH_3_) ppm. HRMS (ESI^+^): *m*/*z* [C_17_H_14_O_6_ + H]^+^ calcd. for [C_17_H_15_O_6_]: 315.0863; found 315.0860.

#### 3.1.6. Synthesis of 3,4-Dimethoxy-9-oxo-9*H*-xanthene-1-carboxylic acid (**11**) and 3,4,6-Trimethoxy-9-oxo-9*H*-xanthene-1-carboxylic acid (**12**)

*3,4-Dimethoxy-9-oxo-9H-xanthene-1-carboxylic acid* (**11**) (20 mg, 71%) was synthesized from 3,4-Dimethoxy-9-oxo-9*H*-xanthene-1-carbaldehyde (**9**) and characterized according to the previously described procedure [20].

To a mixture of 3,4,6-trimethoxy-9-oxo-9*H*-xanthene-1-carbaldehyde (0.176 mmol) in DMF (0.1 M) was added Oxone^®^ (0.176 mmol), and it was stirred under reflux for 24 h. After solvent removal under vacuum, 1N HCl was used to dissolve the salts and EtOAc was added to extract the products. The organic extract was washed with 1N HCl (3x) and brine, dried over Na_2_SO_4_, and the solvent was removed under reduced pressure to obtain the crude product. The product was purified by preparative plates TLC (SiO_2_, EtOAc/MeOH/Formic acid 95:5:1) to give the pure product 3,4,6-trimethoxy-9-oxo-9*H*-xanthene-1-carboxylic acid (**12**, 18 mg, 68%) as white solid.

*3,4,6-Trimethoxy-9-oxo-9H-xanthene-1-carboxylic acid***(12)**; White solid (18 mg, 68% yield); m.p. 175–177 °C. ^1^H-NMR (CDCl_3_, 300.13 MHz): δ = 8.40 (s, 1H, H-2), 8.31 (d, ^3^*J*_8,7_ = 8.8 Hz, 1H, H-8), 7.10–6.89 (m, 2H, H-5, H-7), 4.12 (s, 3H, 3-OCH_3_), 4.10 (s, 3H, 4-OCH_3_), 3.99 (s, 3H, 6-OCH_3_) ppm. ^13^C-NMR (CDCl_3_, 75.47 MHz): δ 178.8 (C-9), 166.9 (C-1′), 165.6 (C-10a), 157.4 (C-6), 156.3 (C-3), 152.1 (C-4a), 140.0 (C-4), 129.2 (C-8), 128.7 (C-1), 117.9 (C-7), 115.5 (C-2), 114.1 (C-9a), 113.8 (C-8a), 99.3 (C-5), 61.9 (3-OCH_3_), 56.9 (4-OCH_3_), 56.4 (6-OCH_3_) ppm. HRMS (ESI^+^): *m*/*z* [C_17_H_14_O_7_ + H]^+^ calcd. for [C_17_H_15_O_7_]: 331.0812; found 331.0810.

#### 3.1.7. General Procedure for the Synthesis of Methyl 3,4-dimethoxy-9-oxo-9*H*-xanthene-1-carbaldehyde (**13**) and Methyl 3,4,6-trimethoxy-9-oxo-9*H*-xanthene-1-carbaldehyde (**14**)

To a mixture of the corresponding methoxy-9-oxo-9*H*-xanthene-1-carbaldehyde (0.176 mmol) in methanol (2 mL) was added Oxone^®^ (0.108 g, 0.176 mmol), and it was stirred under reflux for 5 h. After solvent removal under vacuum, 1N HCl was used to dissolve the salts, and EtOAc was added to extract the products. The organic extract was washed with 1N HCl (3x) and brine, dried over Na_2_SO_4_, and the solvent was removed under reduced pressure to obtain the crude product. Products were purified by preparative plates TLC (SiO_2_, EtOAc/MeOH 9:1) to give the pure products methyl 3,4-dimethoxy-9-oxo-9*H*-xanthene-1-carbaldehyde (**13**, 40.2 mg, 53%) or methyl 3,4,6-trimethoxy-9-oxo-9*H*-xanthene-1-carbaldehyde (**14**, 46.8 mg, 77%) as white solid.

*Methyl 3,4-dimethoxy-9-oxo-9H-xanthene-1-carbaldehyde* (**13**); White solid (40.2 mg, 53% yield); m.p. 169–171 °C. ^1^H-NMR (CDCl_3_, 300.13 MHz): δ = 8.28 (dd, ^3^*J*_8,7_ = 8.8 Hz, ^3^*J*_8,6_ = 1.7 Hz, 1H, H-8), 7.75 (ddd, ^3^*J*_6,5_ = 8.8 Hz, ^3^*J*_6,7_ =7.1 Hz, ^3^*J*_6,8_ = 1.7 Hz, 1H, H-6), 7.59 (d, ^3^*J*_5,6_ = 7.5 Hz, 1H, H-5), 7.40 (ddd, ^3^*J*_7,8_ = 8.1 Hz, ^3^*J*_7,6_ =7.2 Hz, ^3^*J_7_*_,5_ =1.1 Hz, 1H, H-7), 6.99 (s, 1H, H-2), 4.06 (s, 3H, 4-OCH_3_), 4.04 (s, 3H, 3-OCH_3_), ppm. ^13^C-NMR (CDCl_3_, 75.47 MHz): δ = 176.0 (C-9), 170.4 (C-1′), 157.3 (C-10a), 156.3 (C-3), 150.5 (C-4a), 138.0 (C-4), 135.5 (C-6), 130.8 (C-8), 124.9 (C-8a), 121.9 (C7), 118.5 (C-5), 114.3 (C-9a), 108.4 (C-2), 62.3 (3-OCH_3_), 57.2 (4-OCH_3_), 53.8 (COOCH_3_) ppm. HRMS (ESI^+^): *m*/*z* [C_16_H_10_O_5_ + H]^+^ calcd. for [C_16_H_11_O_5_]: 283.06010; found 283.06000.

*Methyl 3,4,6-trimethoxy-9-oxo-9H-xanthene-1-carbaldehyde* (**14**); White solid (46.8 mg, 77% yield); m.p. 169–171 °C. ^1^H-NMR (CDCl_3_, 300.13 MHz): δ = 8.17 (d, ^3^*J*_8,7_ = 8.8 Hz, 1H, H-8), 6.98–6.90 (m, 3H, H-2, H-5, H-7), 4.03 (6H, s, 3-OCH_3_, COOCH_3_), 4.00 (s, 3H, 4-OCH_3_), 3.93 (s, 3H, 6-OCH_3_), ppm. ^13^C-NMR (CDCl_3_, 75.47 MHz): δ = 174.7 (C-9), 170.1 (C-1′), 165.3 (C-6), 157.8 (C-10a), 156.5 (C-3), 150.5 (C-4a), 137.5 (C-4), 129.7 (C-1), 128.3 (C-8), 115.3 (C-8a), 113.9 (C-7, C-9a), 107.7 (C-2), 100.4 (C-5), 61.8 (3-OCH_3_), 56.7 (4-OCH_3_), 56.1 (6-OCH_3_), 53.3 (COOCH_3_) ppm. HRMS (ESI^+^): *m*/*z* [C_18_H_16_O_7_ + H]^+^ calcd. for [C_18_H_17_O_7_]: 345.0969; found 345.0965.

#### 3.1.8. Synthesis of (*Z*)-3,4-Dimethoxy-9-oxo-9*H*-xanthene-1-carbaldehyde oxime (**15**)

To a stirred mixture of 3,4-dimethoxy-9-oxo-9*H*-xanthene-1-carbaldehyde (0.14 mmol) in pyridine (1.48 mL), NH_2_OH.HCl (0.77 mmol) was added. After stirring for 1.5 h at 50 °C, the reaction mixture was evaporated under reduced pressure, and the result residue was extracted with ethyl acetate (3 × 50 mL) from water (20 mL). The resulting organic phase was washed with 5% HCl aqueous (50 mL), water (50 mL), and 10% NaCl aqueous solution (50 mL), dried over Na_2_SO_4_, filtrated, and concentrated to dryness. The crude residue was purified by crystallization from ethyl acetate to give (*Z*)-3,4-dimethoxy-9-oxo-9*H*-xanthene-1-carbaldehyde oxime (**15**) as a white solid.

*(Z)-3,4-Dimethoxy-9-oxo-9H-xanthene-1-carbaldehyde oxime* (**15**); White solid (25.7 mg, 21% yield); m.p. 204–205 °C. ^1^H-NMR (CDCl_3_, 300.13 MHz): δ 11.44 (s, 1H, NOH), 9.31 (s, 1H, H-1′), 8.14 (dd, ^3^*J*_8,7_ = 7.9, ^3^*J*_8,6_ = 1.5, 1H, H-8), 7.85 (ddd, ^3^*J*_6,5_ = 7.5 Hz, ^3^J_6,7_ = 5.8 Hz, ^4^J_6,8_ = 1.7 Hz, 1H, H-6), 7.67 (brdd, ^3^*J*_5,6_= 7.9, 1H, H-5), 7.46 (ddd, ^3^*J*_7,8_ = 8.0 Hz, ^3^*J*_7,6_= 7.5 Hz, ^4^*J*_7,5_ = 1.0 Hz, 1H, H-7), 7.43 (s, H-2), 3.99 (s, 3H, 3-OCH_3_), 3.94 (s, 3H, 4-OCH_3_), ppm. ^13^C-NMR (CDCl_3_, 75.47 MHz): δ = 177.0 (C-9), 156.1 (C-3), 154.7 (C-10a), 150.7 (C-4a), 147.9 (C-1′), 136.9 (C-4), 135.3 (C-6), 129.7 (C-1), 126.1 (C-8), 124.4 (C-7), 121.4 (C-5), 117.8 (C-8a), 113.2 (C-9a), 106.8 (C-2), 61.1 (4-OCH_3_), 56.3 (3-OCH_3_) ppm. HRMS (ESI^+^): *m*/*z* [C_18_H_13_NO_5_ + H]^+^ calcd. for [C_18_H_14_NO_5_]: 300.08665; found 300.08639.

### 3.2. X-Ray Crystallography

Single crystals were mounted on cryoloops using paratone. X-ray diffraction data were collected at 290 K with a Gemini PX Ultra equipped with CuKα radiation (λ = 1.54184 Å). The structures were solved by direct methods using SHELXS-97 and refined with SHELXL-97 [30].

Non-hydrogen atoms were refined anisotropically. Hydrogen atoms were either placed at their idealized positions using appropriate HFIX instructions in SHELXL and included in subsequent refinement cycles or were directly found from difference Fourier maps and were refined freely with isotropic displacement parameters.

Full details of the data collection and refinement and tables of atomic coordinates, bond lengths and angles, and torsion angles have been deposited with the Cambridge Crystallographic Data Centre (CCDC codes below).

#### 3.2.1. Crystal Structure of 1-Methyl-3,4-dimethoxy-9*H*-xanthen-9-one (**1**)

Crystal was monoclinic, space group P21/c, cell volume 1292.06(12) Å^3^ and unit cell dimensions a = 7.2661(4) Å, b = 13.3692(7) Å and c = 13.5206(7) Å, and β = 100.347(5)°. The refinement converged to R (all data) = 5.37% and wR2 (all data) = 13.77%. CCDC 1910877.

#### 3.2.2. Crystal Structure of 1-(Dibromomethyl)-3,4-dimethoxy-9*H*-xanthen-9-one (**7**)

Crystal was monoclinic, space group P21/n, cell volume 1541.66(18) Å^3^ and unit cell dimensions a = 9.7287(7) Å, b = 14.5814(9) Å and c = 11.7638(8) Å, and β = 112.510(8)° (uncertainties in parentheses). The refinement converged to R (all data) = 8.52% and wR2 (all data) = 21.22%. CCDC 1910874.

#### 3.2.3. Crystal Structure of 1-(Dibromomethyl)-3,4,6-trimethoxy-9*H*-xanthen-9-one (**8**)

Crystal was monoclinic, space group P21/n, cell volume 1704.91(8) Å^3^ and unit cell dimensions a = 9.6176(3) Å, b = 17.2880(3) Å and c = 11.0362(3) Å, and β = 111.702(3)°. The refinement converged to R (all data) = 8.59% and wR2 (all data) = 21.46%. CCDC 1910875.

### 3.3. Microbiology

#### 3.3.1. Microorganism Strains and Growth Conditions

In the present study, two Gram-positive bacteria—*Staphylococcus aureus* ATCC 29,213 and *Enterococcus faecalis* ATCC 29212—and two Gram-negative bacteria—*Escherichia coli* ATCC 25,922 and *Pseudomonas aeruginosa* ATCC 27,853—reference strains were used. Multidrug-resistant bacterial strains isolated from public buses (MRSA *S. aureus* 66/1) [31], river water (VRE *E. faecalis* B3/101) [32], and a clinical isolate (ESBL *E. coli* SA/2) were also used when there was a minimum inhibitory concentration (MIC) value for ATCC strains or an inhibition halo. Frozen stocks of all strains were grown in Mueller–Hinton agar (MH – BioKar diagnostics, Allone, France) at 37 °C. All bacterial strains were sub-cultured in MH agar and incubated overnight at 37 °C before each assay. For antifungal activity screening, a yeast reference strain *Candida albicans* ATCC 10231, a filamentous fungi reference strain *Aspergillus fumigatus* ATCC 46645, and tree dermatophyte clinical strains *Trichophyton rubrum* FF5, *Microsporum canis* FF1 and *Epidermophyton floccosum* FF9 were used. Frozen stocks of all fungal strains were sub-cultured in Sabouraud dextrose agar (SDA - BioMérieux, Marcy L’Etoile, France) before each test, to ensure optimal growth conditions and purity.

#### 3.3.2. Antibacterial Susceptibility Testing

An initial screening of the antibacterial activity of the compounds was performed by the disk diffusion method as previously described [16,33]. Briefly, sterile 6 mm blank paper disks (Oxoid, Basingstoke, England) impregnated with 15 µg of each compound were placed on MH agar plates inoculated with the bacteria. A blank disk with DMSO was used as a negative control. MH-inoculated plates were incubated for 18–20 h at 37 °C. At the end of incubation, the inhibition halos where measured. The minimum inhibitory concentration (MIC) was used for determining the antibacterial activity of each compound, in accordance with the recommendations of the Clinical and Laboratory Standards Institute (CLSI) [34]. 10 mg/mL stock solutions of each compound were prepared in dimethylsulfoxide (DMSO – Applichem GmbH, Darmstadt, Germany). For compounds **5**, **10**, **11** and **28**, which were less soluble in DMSO than the other compounds, a stock solution of 2 mg/mL was prepared. In the case of compounds **9**, **12** and **22,** the stock solution prepared was 1 mg/mL. Two-fold serial dilutions of the compounds were prepared in Mueller–Hinton broth 2 (MHB2, Sigma-Aldrich, St. Louis, MO, USA) within the concentration range of 0.062–64 µg/mL. The highest concentration tested was chosen in order to maintain DMSO in-test concentration below 1% (*v*/*v*), as recommended by the CLSI [34]. At this concentration, DMSO did not affect bacterial growth. Cefotaxime (CTX) ranging between 0.031–16 µg/mL was used as a control. Sterility and growth controls were included in each assay. Purity check and colony counts of the inoculum suspensions were also evaluated in order to ensure that the final inoculum density closely approximated the intended number (5 × 10^5^ CFU/mL). The MIC was determined as the lowest concentration at which no visible growth was observed. The minimum bactericidal concentration (MBC) was assessed by spreading 10 µL of culture collected from wells showing no visible growth on MH agar plates. The MBC was determined as the lowest concentration at which no colonies grew after 16–18-h incubation at 37 °C. These assays were performed in duplicate.

#### 3.3.3. Antifungal Activity

The antifungal activity of the test compounds was evaluated against *C. albicans*, *A. fumigatus*, *T. rubrum*, *M. canis* and *E. floccosum*. The MIC of each compound was determined by the broth microdilution method according to CLSI guidelines (reference documents M27-A3 for yeasts [35] and M38-A2 [36] for filamentous fungi). Briefly, cell or spore suspensions were prepared in RPMI-1640 broth medium (Biochrom, Berlin, Germany) supplemented with MOPS (Sigma-Aldrich, St. Louis, MO, USA) from fresh cultures of the different strains of fungi. In the case of filamentous fungi, the inoculum was adjusted to 0.4–5 × 10^4^ CFU/mL for *A. fumigatus* ATCC 46,645 and to 1–3 × 10^3^ CFU/mL for the dermatophytes (*T. rubrum* FF5, *M. canis* FF1 and *E. floccosum* FF9). The inoculum of *C. albicans* was adjusted to 0.5–2.5 × 10^3^ CFU/mL. Two-fold serial dilutions of the compounds were prepared in RPMI-1640 broth medium supplemented with MOPS within the concentration range of 4–128 µg/mL, with maximum DMSO concentration not exceeding 2.5% (*v*/*v*). Sterility and growth controls were also included in each assay. The plates were incubated for 48 h at 35 °C (*C. albicans* and *A. fumigatus*) or during 5–7 days at 25 °C (*T. rubrum*, *M. canis* and *E. floccosum*). MICs were recorded as the lowest concentrations resulting in 100% growth inhibition in comparison to the compound-free controls. Voriconazole MIC for *Candida krusei* ATCC 6258 was determined as quality control [35,36]. The results obtained were within the recommended limits. The minimum fungicidal concentration (MFC) was determined by spreading 20 µL of culture collected from wells showing no visible growth on SDA plates. The MFC was determined as the lowest concentration showing 100% growth inhibition after 48 h at 35 °C (for *C. albicans* and *A. fumigatus*) or 5–7 days incubation at 25 °C (*T. rubrum*, *M. canis* and *E. floccosum*). All the experiments were repeated independently at least two times.

#### 3.3.4. Antibiotic Synergy Testing

In order to evaluate the combined effect of the compounds and clinically relevant antimicrobial drugs, a screening was conducted using the disk diffusion method, as previously described [16,33]. A set of antibiotic disks (Oxoid, Basingstoke, England) to which the isolates were resistant was selected: cefotaxime (CTX, 30 µg) for *E. coli* SA/2, oxacillin (OX, 1 µg) for *S. aureus* 66/1, and vancomycin (VA, 30 µg) for *E. faecalis* B3/101. Antibiotic disks alone (controls) and antibiotic disks impregnated with 15 µg of each compound were placed on MH agar plates seeded with the respective bacteria. Sterile 6-mm blank paper, impregnated with 15 µg of each compound, alone, was also tested. A blank disk with DMSO was used as a negative control. MH-inoculated plates were incubated for 18–20 h at 37 °C. Potential synergism was recorded when the halo of an antibiotic disk impregnated with a compound was greater than the halo of the antibiotic or compound-impregnated blank disk alone.

#### 3.3.5. Antifungal Synergy Testing

In order to evaluate the combined effect of the compounds and clinically relevant antifungal drugs, a checkerboard assay was conducted, as previously described [37]. Fluconazole was used in a range between (0.0625–4 µg/mL), and compounds were tested in a range between their MIC and progressive two-fold dilutions. Potential synergism was recorded when the inhibitions of the combined compounds with antifungals was greater than the compounds or the antifungals alone. Fractional inhibitory concentration (FIC) was calculated as follows: FIC of compound = MIC of compound in combination with antifungal/MIC of compound alone, and FIC of antifungal = MIC of antifungal in combination with compound/MIC of antifungal alone. FIC index (∑FIC) = FIC of compound + FIC of antifungal. ∑FIC ≤ 0.5 indicates synergy, 0.5 < ∑FIC ≤ 4 indicates no interaction, ∑FIC > 4 indicates antagonism.

#### 3.3.6. Germ Tube Inhibition Assay

The effect of compounds **3**, **4**, **13**, **23**, **24**, **26**, **27** and **31** in germ tube formation of *C. albicans* was determined as follows: cell suspensions from overnight SDA cultures of *C. albicans* ATCC 10,231 were prepared in NYP medium (*N*-acetylglucosamine [Sigma, St. Louis, MO, USA; 10^−3^ mol/L], yeast nitrogen base [Difco, New Jersey, USA; 3.35 g/L], proline [Fluka, Buchs, St. Gallen, Switzerland; 10^−3^ mol/L], and NaCl [4.5 g/L], pH 6.7 ± 0.1) and adjusted to a density of (1.0 ± 0.2) × 10^6^ CFU/mL, as determined by cell counts using a hemocytometer. An appropriate volume of compound stock solution at 10 mg/mL was added in order to obtain final concentrations ranging between 128 and 16 μg/mL. Filamentation controls were included in each assay with and without 1.28% DMSO. Following a 3-h incubation at 37 °C, 100 cells from each sample were counted, using a hemocytometer, and the percentage of germ tubes was determined. Germ tubes were considered when they were at least as long as the blastospore. Protuberances showing a constriction at the point of connection to the mother cell, typical for pseudohyphae, were not considered [38]. At least three independent assays were performed.

#### 3.3.7. Biofilm Formation Inhibition Assay

The effect of **3** and **27** on biofilm formation was evaluated through quantification of total biomass by crystal violet staining [26,39]. Briefly, each compound, in concentrations ranging between 128 and 16 μg/mL, was added to *C. albicans* ATCC 10,231 suspensions prepared in RPMI-1640 broth medium supplemented with MOPS, from overnight SDA cultures, at a final concentration of (1.0 ± 0.2) × 10^6^ CFU/mL, as determined by cell counts using a hemocytometer. A control with appropriate concentration of DMSO, as well as a negative control (RPMI-1640 alone), was included. Sterile 96-well flat-bottomed untreated polystyrene microtiter plates were used. After a 48-h incubation at 37 °C, the biofilms were stained with 1% (*v*/*v*) crystal violet (Química Clínica Aplicada, Amposta, Spain) for 5 min. The stain was solubilized with 33% (*v*/*v*) acetic acid (Acetic acid 100%, AppliChem, Darmstadt, Germany), and the biofilm biomass was quantified by measuring the absorbance of each sample at 570 nm in a microplate reader (Thermo Scientific Multiskan^®^ EX, Thermo Fisher Scientific, Waltham, MA, USA). The background absorbance (RPMI-1640 without inoculum) was subtracted from the absorbance of each sample and the data are presented as percentage of control. Three independent assays were performed in triplicate for each experimental condition.

## 4. Conclusions

Using several straightforward transformations, it was possible to synthetize a series of structurally diverse xanthones (**3**–**15**) and evaluate their antimicrobial activity, along with a series of 19 in-house 3,4-dioxygenated xanthones **16–33**. The results of antimicrobial screening revealed the potential of some of the obtained compounds either as new antibacterial (**7, 8, 12, 20, 26** and **27**) or antifungal (**3**, **26** and **27**) agents, with compound **3** exhibiting a potent inhibitory effect on the growth of dermatophyte clinical strains (*T. rubrum* FF5, *M. canis* FF1 and *E. floccosum* FF9), with a MIC of 16 µg/mL for all the tested strains. Compounds **3** and **26** also showed a potent inhibitory effect of *C. albicans* ATCC 10,231 germ tube and biofilm formation, important virulence factors. In general, it was observed that compounds with halogen atoms were the most promising in terms of antibacterial activity, with some of the representatives being highly active against either Gram-positive or Gram-negative strains. However, although some of the aminated xanthones exhibited low solubility, their conversion to hydrochloride salts increase their potential as antibacterial or antifungal agents.

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
