# Peer review of "Synthesis of a Small Library of Nature-Inspired Xanthones and Study of Their Antimicrobial Activity"

_molecules, 2020, doi:10.3390/molecules25102405_

Round 1
Reviewer 1 Report
The Authors presented results on synthesis, characterisation of novel xanthones followed by their initial antibacterial investigations. The manuscript is well - written and deserves publication in Molecules. However, the Aurhors may comment low yield in the case of oxime 15. Is it problem of isolation or selectivity of nucleophilic addition?
Author Response
The Authors presented results on synthesis, characterisation of novel xanthones followed by their initial antibacterial investigations. The manuscript is well - written and deserves publication in Molecules. However, the Aurhors may comment low yield in the case of oxime 15. Is it problem of isolation or selectivity of nucleophilic addition?
Reply: We thank reviewer #1 for his/her valuable comments, and in order to justify the low yields obtained in the synthesis of oxime 15 the following sentence was added in the revised manuscript “Furthermore, a condensation of the xanthone 9 with hydroxylamine allowed to obtain the aldoxime 15 with a moderate yield (21%), justified by laborious purification and consequent product losses.”.
Reviewer 2 Report
This manuscript describes the synthesis of xanthone-type compounds and the evaluation of their biological activities. The authors prepared a series of thirteen xanthones derivatives (compounds 3-15) from two simple oxygenated xanthones. Then, they evaluated antimicrobial activity of these xanthones derivatives (compounds 3-15) along with previously synthesized aminated xanthones (compounds 16-33). They found that some of the compounds showed a potent inhibitory effect. It was revealed that compounds containing halogen atoms are the most promising in terms of antibacterial activity. Overall, the work is well executed and described. There are a few minor suggestions:
1. The authors can discuss about yield in their chemical synthesis. For example, the authors could explain why the conversion from compound 2 to compound 6 was in low yield.
2. In Table 2, 13C signals for methoxy group can be included.
3. The structures of compounds 16-33 are not shown in this manuscript. Table 3 shows only R1 and R2 substituents. As for compounds 18-33, R1 is missing. Compounds 18-28 seem to be the same, judging from Table 3. Therefore, the authors should provide the chemical structures of compounds 16-33.
Author Response
This manuscript describes the synthesis of xanthone-type compounds and the evaluation of their biological activities. The authors prepared a series of thirteen xanthones derivatives (compounds 3-15) from two simple oxygenated xanthones. Then, they evaluated antimicrobial activity of these xanthones derivatives (compounds 3-15) along with previously synthesized aminated xanthones (compounds 16-33). They found that some of the compounds showed a potent inhibitory effect. It was revealed that compounds containing halogen atoms are the most promising in terms of antibacterial activity. Overall, the work is well executed and described. There are a few minor suggestions:
- The authors can discuss about yield in their chemical synthesis. For example, the authors could explain why the conversion from compound 2 to compound 6 was in low yield.
We thank reviewer #2 for his/her valuable comments, and the sentence “The low yields obtained in this reaction are due to failure of the reaction to go to completion and other complications on the purification step.” was added to the manuscript in order to justify the low yields obtained in the synthesis of compounds 5 and 6.
To justify the low yields obtained in the synthesis of oxime 15 the following sentence was added in the revised manuscript “Furthermore, a condensation of the xanthone 9 with hydroxylamine allowed to obtain the aldoxime 15 with a moderate yield (21%), justified by laborious purification and consequent product losses.”.
- In Table 2, 13C signals for methoxy group can be included.
Reply: As suggested by reviewer #2, the 13C signal values for methoxy groups were added to Table 2.
- The structures of compounds 16-33 are not shown in this manuscript. Table 3 shows only R1 and R2 substituents. As for compounds 18-33, R1 is missing. Compounds 18-28 seem to be the same, judging from Table 3. Therefore, the authors should provide the chemical structures of compounds 16-33.
Reply: We thank reviewer #2 for his/her comment. To clarify this aspect, additional lines were added to the table to make the separation between the structures more visible. However, we think that compounds 16-33 are well represented in Table 3. If R1 and R2 are substituted in the general structure that is represented on the top of the table, the structures of compounds 16-33 will be obtained. We would also like to point out that some of the compounds have R2 = OH (instead of OCH3) and others are the corresponding salts (for example, 28 is the corresponding salt of 19).